# Integrated ATAC-Seq and RNA-Seq Data Analysis to Reveal *OsbZIP14* Function in Rice in Response to Heat Stress

**DOI:** 10.3390/ijms24065619

**Published:** 2023-03-15

**Authors:** Fuxiang Qiu, Yingjie Zheng, Yao Lin, Samuel Tareke Woldegiorgis, Shichang Xu, Changqing Feng, Guanpeng Huang, Huiling Shen, Yinying Xu, Manegdebwaoga Arthur Fabrice Kabore, Yufang Ai, Wei Liu, Huaqin He

**Affiliations:** College of Life Sciences, Fujian Agriculture and Forestry University, Fuzhou 350002, China; 1200514046@fafu.edu.cn (F.Q.); 1200525042@fafu.edu.cn (Y.Z.); 3205403002@stu.fafu.edu.cn (Y.L.); samuel_tareke@163.com (S.T.W.); xushichang@fafu.edu.cn (S.X.); 2210525004@fafu.edu.cn (C.F.); 1210525015@fafu.edu.cn (G.H.); 1210514066@fafu.edu.cn (H.S.); 1210514101@fafu.edu.cn (Y.X.); 2211925002@fafu.edu.cn (M.A.F.K.); aiyufang@fafu.edu.cn (Y.A.); weilau@fafu.edu.cn (W.L.)

**Keywords:** *Oryza sativa* L., transcription factors, heat stress, *OsbZIP14*

## Abstract

Transcription factors (TFs) play critical roles in mediating the plant response to various abiotic stresses, particularly heat stress. Plants respond to elevated temperatures by modulating the expression of genes involved in diverse metabolic pathways, a regulatory process primarily governed by multiple TFs in a networked configuration. Many TFs, such as WRKY, MYB, NAC, bZIP, zinc finger protein, AP2/ERF, DREB, ERF, bHLH, and brassinosteroids, are associated with heat shock factor (Hsf) families, and are involved in heat stress tolerance. These TFs hold the potential to control multiple genes, which makes them ideal targets for enhancing the heat stress tolerance of crop plants. Despite their immense importance, only a small number of heat-stress-responsive TFs have been identified in rice. The molecular mechanisms underpinning the role of TFs in rice adaptation to heat stress still need to be researched. This study identified three TF genes, including *OsbZIP14*, *OsMYB2*, and *OsHSF7*, by integrating transcriptomic and epigenetic sequencing data analysis of rice in response to heat stress. Through comprehensive bioinformatics analysis, we demonstrated that *OsbZIP14*, one of the key heat-responsive TF genes, contained a basic-leucine zipper domain and primarily functioned as a nuclear TF with transcriptional activation capability. By knocking out the *OsbZIP14* gene in the rice cultivar Zhonghua 11, we observed that the knockout mutant *OsbZIP14* exhibited dwarfism with reduced tiller during the grain-filling stage. Under high-temperature treatment, it was also demonstrated that in the *OsbZIP14* mutant, the expression of the *OsbZIP58* gene, a key regulator of rice seed storage protein (SSP) accumulation, was upregulated. Furthermore, bimolecular fluorescence complementation (BiFC) experiments uncovered a direct interaction between *OsbZIP14* and *OsbZIP58*. Our results suggested that *OsbZIP14* acts as a key TF gene through the concerted action of *OsbZIP58* and *OsbZIP14* during rice filling under heat stress. These findings provide good candidate genes for genetic improvement of rice but also offer valuable scientific insights into the mechanism of heat tolerance stress in rice.

## 1. Introduction

Rice is one of the world’s most important food crops, and its productivity and quality are vital for global food security [1]. With global warming and the increasing frequency of hot weather, rice productivity is subjected to more extreme temperature stress than ever before [2]. In southern China, rice-producing areas are significantly impacted by heat and drought, which limit rice productivity [3]. Rice is highly sensitive to heat stress, especially during the reproductive and grain-filling stages. Heat adversely affects rice quality, such as increasing amylose, decreasing kernel weight, and increasing chalkiness [4]. The physiological deterioration of quality is mainly due to an accelerated grain-filling rate and the grain-filling period being shortened under heat-stress conditions [5,6,7]. The transcriptional regulation of genes involved in storage protein biosynthesis is crucial for rice quality under heat conditions [8,9]. Therefore, understanding molecular mechanisms by which plants perceive and transduce stress signals to cellular machinery to initiate adaptive transcription factors for heat stress tolerance in plants is an essential prerequisite for identifying key genes and pathways to breed heat-stress-tolerant crops [10].

Transcription factors (TFs) in plants play crucial roles in response to a range of abiotic stresses. They are continuously synthesized in plants and relay and amplify stress signals within cells. TFs can precisely interact with cis-acting elements in gene promoter regions, thereby activating or inhibiting the transcription of specific genes [11]. Furthermore, they induce changes in plant physiology and biochemistry by regulating the expression of downstream-related genes [12]. There are more than 50 known TF families in plants, which often have the characteristic of simultaneously regulating multiple stress responses due to more complex internal connection between various TFs [13]. Among them, heat shock transcription factors (HSFs), dehydration response element binding protein (DREB), multi-protein binding factor (MBF), WRKY, MYB, ERF, NAC, and basic leucine zipper (bZIP) have been involved in heat stress response [14,15,16,17].

Transcriptome sequencing (RNA-seq) is a powerful tool for understanding plant gene expression and screening for stress-resistance genes [18,19,20]. However, as high-throughput sequencing technology advances, many omics technologies emerge. Any mono-omics is insufficient to explain the mechanism of plant stress response systematically, and the combination of multi-omics is an inevitable trend of future development [21]. The combination of epigenome and transcriptome analysis has attracted more and more attention. Assay for Transposase Accessible Chromatin with High-Throughput Sequencing (ATAC-Seq) is a widely used technique for determining chromatin accessibility throughout the genome. This method is sensitive, specific and uses fewer cells [22,23]. Due to the simple procedure and low material input, ATAC-seq has been widely used in the detection of DNA regulatory regions in plants such as Arabidopsis [24], rice [25], tomato [26], and wheat [27]. A study focused on the response of grapes to cold stress identified nine cold-responsive TFs, including *CBF4*, *RAV1*, and *ERF104*, by integrating ATAC-seq and RNA-seq methods [28]. Similar methods were also applied to identify nine TF genes including ATHB7, HAT5, and WRKY26 in apple that played important roles in drought resistance [29]. Therefore, the combined analysis of ATAC-seq and RNA-Seq is a powerful tool in rapidly and efficiently identifying stress-responsive TFs in plants.

To the best of our knowledge, only a few TF genes have been reported which play key roles in rice in response to heat stress. However, the mechanism of how different TFs respond to heat stress is still unclear. In this study, the transcriptomic and epigenomic data of rice under heat stress were used to screen the heat-responsive TF genes. Then we used different protein-protein docking tools to predict the interaction between the predicted TF gene and a target gene. Finally, we verified the interaction by using the Bimolecular Fluorescence Complementation (BiFC) method. The results of this study provide potential heat stress tolerant candidate genes and provide scientific inspiration for understanding the heat tolerance of rice.

## 2. Results

### 2.1. Analysis of the Chromatin Accessibility in Rice under Heat Stress

To define the chromatin accessibility of rice under heat stress, ATAC-seq data analysis was performed. Signals of open chromatin were obtained by alignment, removing duplication, and peak calling. The result showed consistent peaks in both the control and treated, confirming the reproducibility of the ATAC-seq libraries used in this study (Appendix A). Structural annotation of the peaks showed that the ratio of ATAC signal transposase hypersensitive sites (THSs) present in the promoter region (0–3 kb upstream of transcription start site (TSS)) in rice under heat stress was 68.25% (Figure 1A), while it was 66.60% in rice under control condition (Figure 1A). This indicated that the THS in the promoter region slightly increased in rice under heat stress. Heatmap and signal average plot analysis of the THSs demonstrated that heat stress promoted signaling at the TSS and enhanced accessibility in the transcription initiation region (Figure 1C). We then characterized the number of THSs used to identify a gene. We found that most of the THSs were uniquely mapped to one gene. However, the ratio of THSs mapped to one gene decreased after heat stress treatment, whereas the proportion of THSs corresponding to two or three genes showed a relative increment (Figure 1B).

### 2.2. Analysis of the Enriched Motif in ATAC Peaks in Rice under Heat Stress

THS enrichment was performed by using Homer software. The enrichment results showed that 1652 THSs were positively enriched whereas 156 THSs were negatively enriched, of which differentially enriched THSs accounted for 7.07% of the overall THSs identified (Appendix A). Structural annotation of these differentially enriched THSs showed that the proportions of positive and negative enrichment regions in the upstream 0–1, 1–2, 2–3 Kb regions of TSS were 34.18% and 48.39%, 17.68 and 19.35%, 13.07%, and 10.32%, respectively, and the proportions in the intergenic regions were 33.41% and 17.42%, respectively. The highest proportion at the genomic level was in the upstream ≤1 Kb region of TSS (Appendix A). This indicated that heat-responsive cis-regulatory elements were primarily located in the core promoter.

To identify the genes with heat-responsive binding elements, we used the TFs in the model plant Arabidopsis to search in the rice genome with the enriched motifs (Appendix A). This analysis led to the identification of 37 rice genes with heat-responsive TFs (Appendix A). After removing duplicate genes and selecting the genes with maximum identity (e-value < 1 × 10^−5^), we identified five TF genes, including OsbZIP14, OsMYB2, OsHsfB2b, and OsMYB30 and OsHSF7 (Table 1).

### 2.3. Identification of DEGs in Rice under Heat Stress

High-throughput mRNA sequencing data of rice under 36 h heat stress was used to evaluate the transcriptome. In total, 24283 genes were identified in both the heat stress treated and control samples. To identify the critical functional genes related to heat stress resistance, differentially expressed genes (DEGs) were filtered based on the criteria |log2(fold change)| ≥ 1 and *p*-Value < 0.05. Overall, 1942 upregulated and 3617 downregulated genes were screened (Figure 2). This result indicated that plant life activity was somewhat suppressed under heat stress.

To determine whether changes in open chromatin regions were associated with changes in gene expression in rice under heat stress, we integrated ATAC-seq data with RNA-seq data analysis. We found that among the five heat-responsive key TF genes identified in ATAC-seq data analysis, three were significantly regulated by heat stress. Specifically, OsbZIP14 (bZIP2) was 2.30-fold down-regulated, OsHSFA1E (OsHSF7) was 44.6-fold downregulated, and OsMYB2 (MYB108) was 25.0-fold down-regulated (Figure 3 and Table 2). 

### 2.4. Verification of the Gene Expression of the Key TF Genes in Rice under Heat Stress

We verified the expression of three heat-responsive TFs, including OsbZIP14, OsHSF7, and OsMYB30, by using RT-qPCR. As shown in Figure 4, the expression of these genes was 2.91, 2.49, and 1.41-fold downregulated, respectively, under heat stress compared to the control. This was consistent with the results in RNA-seq data analysis. 

### 2.5. Sub-Cellular Localization and Transcriptional Activity of OsbZIP14

To determine the transcription factor function of OsbZIP14, we conducted the subcellular localization analysis of OsbZIP14 gene. We used pUC-OsbZIP14-GFP and pCHF3-OsbZIP14-EGFP vectors to transform rice protoplast cells and tobacco epidermal cells, respectively. The resulting fusion proteins were then observed by fluorescence microscopy. The signal from both rice protoplast cells and tobacco epidermal cells showed that the OsbZIP14 protein was localized in the nucleus, as shown in Figure 5A and Figure 5B. After that, we used the yeast two-hybrid technique to verify the self-activation and transcriptional activation effect of OsbZIP14. The positive control bacterial solution for interaction grew normally on both the two-deficient and four-deficient media, while the negative control did not grow on the four-deficient medium (Figure 5D). Co-transformation of the OsbZIP14 gene product protein and the corresponding empty vector confirmed that OsbZIP14 has a self-activation function and transcriptional activation activity. These wet experiment results confirmed that OsbZIP14 was a transcription factor with a transcriptional activation function.

### 2.6. Interacting Proteins of OsbZIP14

The yeast two-hybrid library was developed and used to identify the proteins that interact with OsbZIP14 in rice under heat stress. Inhibition of OsbZIP14 yeast growth in three-deficient and four-deficient media was observed with 3-AT and Aureobasidin A (AbA) tobacco at 5.0 and 20.0 mmol, respectively (Appendix A). The PGBKT7-OsbZIP14 plasmid was transformed into yeast cells and then transformed into the rice library plasmid. A single colony was selected and grown in three deficient media, followed by spreading on four deficient media with X-α-Gal to observe reporter gene expression (Appendix A). Positive clones were selected and analyzed by PCR, and 13 single bands (Appendix A), accounting for 54.1% of the total, were obtained. Sequencing of the 13 bands showed that eight bands were successfully sequenced, among which five were annotated as CCR4-NOT transcription complex subunit I, photosystem I assembly protein, NLS receptor mRNA, bZIP transcription factor, and protein PELPK1 (Appendix A). 

To confirm the interaction between OsbZIP14 and a potential interacting protein from the yeast two-hybrid library, molecular docking was performed using HDOCK software (Table 3). The model with the lowest energy, Model1, was selected and analyzed using PLIP and PDBsum tools to predict the interaction of proteins. We found a hydrogen bond between ASN74 of OsbZIP58 and GLU274 of OsbZIP14, suggesting the possible interaction between the two proteins (Figure 6). PDBsum tool also showed that there was a salt bridge, three hydrogen bonds, and between 31 interfacing amino acid residues in Q6ZLB0 (OsbZIP58 protein) and 28 interfacing amino acid residues in Q0E476 (OsbZIP14 protein) (Figure 7). Based on the surface area analysis, the total surface area of these two proteins was 1924:2026, indicating the possibility of their interaction.

To further confirm the direct interaction between OsbZIP14 and OsbZIP58 products, the ORFs of OsbZIP14 and OsbZIP58 were cloned by PCR to generate the vectors N-OsbZIP58/C, N/C-OsbZIP14, and N-OsbZIP58/C-OsbZIP14 (Figure 8). The control was a mixture of C-OsbZIP14, N-OsbZIP58, and the corresponding empty plasmids, while the treatment group was a mixture of the pRTVnVC-OsbZIP14 and pRTVnVN-OsbZIP58 plasmids. The corresponding plasmids were then transferred into rice protoplasts, and the results were observed with a laser confocal system after 9 h culture. The red and green lights confirmed that both plasmids were successfully transferred into the protoplasts. The yellow light was observed in the treated samples, indicating a direct interaction between OsbZIP14 and OsbZIP58 proteins in rice, whereas the control group did not show any yellow light.

### 2.7. Validation of the Functions of OsbZIP14 Gene in Rice

The over-expression (OE) of the OsbZIP14 gene and its wild-type (WT) rice seedlings were treated at 45 °C for 36 h. After treatment, it was observed that the leaf tip of WT and OE plants withered, yellowed, and aged, but the OE lines showed better growth than WT (Figure 9A). After this, DAB (3, 3‘-diaminobenzidine) and trypan blue staining were used to detect H_2_O_2_ accumulation and cell death of rice seedlings under heat stress (Figure 9C). It was found that under heat stress, the leaf tip of the WT lines was stained deeper than that of the OE lines, indicating that the accumulation of H_2_O_2_ and the level of cell death in WT were higher than those of OE lines under heat stress. To further investigate the impact of the heat treatment, we transferred the OE and WT plants to normal condition (28 °C) for one month to recover from stress. It was demonstrated that the survival rate of OsbZIP14 OE and its WT lines were 65.22% and 49.60%, respectively (Figure 9D). On the other hand, these results implied that over-expression of OsbZIP14 gene contributed to the plants’ recovery from rice heat stress (Figure 9B). 

To further investigate the role of OsbZIP14 in rice in response to heat stress, we compared the growth and development of OsbZIP14 gene knockout (KO) and its WT rice plants during the grain-filling stage under heat stress (Appendix A). The results showed that the OsbZIP14 mutants were dwarfs with fewer tillers under heat stress when compared to the WT plants, indicating that OsbZIP14 promoted the growth and development of rice plants under heat stress (Figure 10A). After exposing the mutant and WT rice plants at a grain-filling stage to 38 °C heat stress for 5d, we found that the relative expression level of OsbZIP58 in the OsbZIP14 mutant plants was significantly increased (Figure 10B), suggesting that OsbZIP14 had a negative regulatory effect on the expression of OsbZIP58.

## 3. Discussion

Heat stress poses a major abiotic challenge to the growth and productivity of rice in many countries worldwide. Understanding the adaptive mechanisms of rice to heat stress may facilitate the development of heat-tolerant rice varieties, thereby enhancing productivity in regions with warm climates. he alterations in the expression levels and patterns of TF genes may have a significant impact on rice’s ability to adapt to various abiotic stressors, including light, water, and temperature. With the current focus on climate resilience, a closer examination of the diverse TF genes can lead to more precise crop improvement breeding. Moreover, advances in sequencing technology and omics resources, such as transcriptomics, proteomics, and epigenome sequencing, are making it increasingly feasible to identify stress-responsive TFs. In this study, we combined RNA-seq and ATAC-seq data analysis to identify novel TFs that might play key roles in heat stress responses in rice, along with studying their adaptive mechanisms for heat stress. 

Given the correlation between TF regulation and chromatin accessibility, we sought to identify differentially expressed TFs and motifs enriched in differential chromatin accessibility regions (DARs). Three TF genes were identified, including OsHSF7, OsbZIP14, and OsMYB2. Among these, OsHSF7 had been found to play a critical role in the heat stress responsive pathways in rice. Liu et al. (2009) found that this gene was rapidly expressed in high levels in response to temperature, which indicates that it may be involved in heat stress reception and response [30]. In another study, Cheng et al. (2014) also reported that OsHSF7 encodes two main splice variant proteins, OsHSFA2dI and OsHSFA2dII in rice. Under normal conditions, OsHSFA2dII is dominant but it is a transcriptionally inactive spliced form. However, when the plant is under heat stress, OsHSFA2d is alternatively spliced into a transcriptionally active form, OsHSFA2dI, which participates in the heat stress response [31]. OsMYB2 was also identified as a regulator in cold and salt stress responses in rice [32]. Similarly, OsbZIP14 was also reported to be involved in drought stress tolerance in rice [33]. All these results indicated that the three TFs identified in this study had been reported to be involved in rice’s response to different abiotic stresses. Several reports have shown that the expression of bZIP transcription factor genes is altered in response to various abiotic stresses [34,35]. Thus, in this study, we primarily focused on the gene, OsbZIP14, to further investigate its role in rice in response to heat stress.

In this investigation, we successfully amplified the crucial TF gene, OsbZIP14, whose expression was significantly downregulated by heat stress. We found that OsbZIP14 was a nuclear transcription factor with transcriptional activation ability and which is localized in the nucleus. To further confirm its function in heat stress response, we obtained homozygous mutant materials for OsbZIP14 loss-of-function and OE through continuous multi-generational breeding in the field. The phenotypic and physiological studies showed that OsbZIP14 played a role in plant growth and development under heat stress. Many senescence-associated mutants have been identified in various plant species to interpret the senescence-related genetic mechanisms [36,37]. To explore the association between the OsbZIP14 gene and heat stress-induced premature leaf senescence, we carried out a heat treatment on OsbZIP14 OE lines at the seedling stage. Excessive accumulation of ROS (reactive oxygen species) such as hydrogen peroxide (H2O2) can induce plant cell death [38,39]. We hence detected H_2_O_2_ accumulation and cell death of OsbZIP14 OE seedlings with and without heat treatment by using 3,3′- diaminobenzidine (DAB) and trypan blue staining. The results showed that OsbZIP14 OE had a similar level of H_2_O_2_ accumulation and cell death before heat treatment at 28 °C, whereas WT had higher levels of H_2_O_2_ accumulation and cell death after heat treatment at 45 °C compared with OsbZIP14 OE (Figure 9C). These results indicated that OsbZIP14 could positively contribute to heat tolerance, while dysfunction of OsbZIP14 in WT resulted in higher heat stress susceptibility associated with cell membrane damage and H_2_O_2_-induced cell death. Previous studies have also found that the bZIP TFs played pivotal roles in the plant’s abiotic stress response [40]. In *Arabidopsis thaliana*, bZIP1 was known as a key TF implicated in light and nitrogen sensing [41], while OsbZIP60 was a significant regulator of rice chalkiness and an essential player in water conservation and heat injury resistance [15,42].

Furthermore, the computational analysis and wet experimental results demonstrated that OsbZIP14 protein could directly interact with OsbZIP58 protein product, in which the expression was highly upregulated in the OsbZIP14 mutants at grain filling stage under heat stress (Figure 9C). In silico protein docking analysis was applied through a template based modeling to obtain the lowest energy model, which was a critical step for the subsequent analysis in which PLIP and PDBsum software predicted the structural interaction between OsbZIP14 and OsbZIP58 [43,44,45]. Both tools predicted the presence of hydrogen bonds between the two interacting proteins. The hydrogen bond formation provides a protein and its ligands a directionality and specificity of interaction that is a fundamental aspect of molecular recognition [46]. In another study by Jiang et al.(2002), it was suggested that the hydrogen bond makes an important contribution to the association of stability of protein complexes and in protein–protein interaction studies [47]. Besides the hydrogen bond, the PDB-sum tool also predicted the presence of three salt bridges between the two genes. Salt bridges are also known for their preferences within proteins and at protein–protein interfaces [48]. Finally, the specific interactions of these two genes were confirmed by both pairwise Y2H and BiFC assays.

The bZIP family members have been found to be engaged in cooperative binding as homodimers to achieve synergistic effects and interactions [49]. Meanwhile, It was reported that OsbZIP58 and Dof transcriptional activator RPBF jointly regulated the expression of seed storage proteins and grain filling of rice [50,51]. A study conducted by Xu et al. (2020) also revealed that heat stress caused selective splicing of the OsbZIP58 gene, which affects the accumulation of storage substances and grain quality. According to their finding, under heat stress alternate splicing of OsbZIP58 results in two protein isoforms, the full length OsbZIP58a and the shorter one OsbZIP58b with lower activity. It was also demonstrated that the efficiency of OsbZIP58 pre-mRNA splicing was higher in more thermo-tolerant rice varieties, suggesting a direct link between pre-mRNA splicing and heat stress seed resilience [51]. Despite the phenotypic variation, our findings demonstrated that the OsbZIP58 gene was up-regulated both in the wild type and the mutant. Therefore, OsbZIP14 might play a key role in efficient pre-mRNA splicing of OsbZIP58 to regulate its expression at the grain filling stage of rice under heat stress.

## 4. Materials and Methods

### 4.1. Data Collection Procedures

The rice reference genome IRGSP-1.0_genome and the rice genome gff annotation file were downloaded from the RAP-DB (Rice Annotation Project Database) (https://rapdb.dna.affrc.go.jp/download/irgsp1.html, accessed 10 September 2021). The RNA-seq and ATAC-seq data of rice under heat stress were collected from the NCBI-SRA database (https://www.ncbi.nlm.nih/gov/sra/, accessed 1 October 2021) (Appendix A).

### 4.2. RNA-seq Data Analysis

We collected the Nipponbare rice RNA-seq data (accession ID: PRJNA604026) under heat stress treatment from the NCBI archive. Then clean data was obtained by removing adapter linkers and low-quality sequences using Trim Galore software v0.6.5 (http://www.bioinformatics.babraham.ac.uk/projects/trim_galore/ accessed on 23 October 2021). Clean reads were then mapped to the reference genome using HISAT2 v2.2.1 (accessed on 25 October 2021) [52,53], sorted using samtools v1.9 (accessed on 25 October 2021) [54], and FPKM (fragments per kilobase of exon model per million mapped fragments) was calculated for each gene. Finally, differential expression analysis was performed on the 6 sample expression matrix using the R v4.0.3 DESeq2 package v1.30 (accessed on 28 October 2021) [55]. The significance level of each gene was determined based on corrected *p*-Value (padj < 0.05) and expression fold change |log2 (fold change)| > 1. 

### 4.3. ATAC-seq Data Analysis

We collected the Nipponbare rice ATAC-seq data (accession ID: PRJNA604028) under heat stress treatment from NCBI archive. Using a similar approach to the RNA-seq analysis, data quality control, filtering, and comparison were performed. After this, uniquely aligned sequence fragments were identified and selected for further analysis. The MACS2 (accessed 3 October 2021) was used to call peaks, using a *p*-Value threshold of 0.01 and default settings. R v4.0.3 ChIPseeker package v1.30.2 (accessed 5 October 2021) was used for peak annotation, and peak differences between experimental and control groups were analyzed using the R v4.0.3 Diffbind package v2.12 (accessed 6 October 2021). Motifs corresponding to differences in peaks were identified using Homer software v4.11 (accessed 7 October 2021). Finally, the rice genes with enriched thermo-sensitive motifs were identified by using blast search against known Arabidopsis thaliana TF genes.

### 4.4. Experimental Design and Procedures

Rice cultivars Nipponbare and ZH11 were used as materials in this study. Nipponbare *OsbZIP14* over-expression (OE) and wildtype (WT) seedlings were grown in a light culture room under controlled environmental conditions of 28 °C and a photoperiod of 14h light/10 h dark until they were 12 days old. To impose heat stress, rice seedlings at three leaves stage were subjected to 45 °C (treatment) or 28 °C (control) conditions with 85% relative humidity and 10,000 lux light in a light-temperature incubator for 36 hrs. Then normal growth condition was resumed for one month to measure the survival rate of the rice seedlings, whilst to assess the function of the *OsbZIP14* TF gene in rice growth under heat stress, ZH11 *OsbZIP14* knockout (KO) and wild-type (WT) rice plants when grain filling during the early ripening period were exposed to 38 °C (treatment) and 25 °C (control) for a 14 h light/10 h dark photoperiod, 80.5% relative humidity, and 10,000 lux light intensity. Following these treatments, rice leaves or grains were sampled, frozen immediately in liquid nitrogen, and stored at −80 °C for further analysis.

### 4.5. RT-qPCR Analysis

Total RNA was extracted from rice leaves using the MiniBEST Plant RNA Extraction Kit (TaKaRa, Beijing, China), and cDNA was synthesized using the PrimeScript™ RT reagent Kit (TaKaRa). Real-time quantitative PCR (qRT-PCR) was performed on a 96-well plate using the Bio-Rad CFX96 Real-Time PCR Detection System (Bio-Rad) and TB Green^®^ Premix Ex Taq™ II (Tli RnaseH Plus) (TaKaRa). The rice actin gene was used as the reference gene to normalize the target gene expression, which was calculated using the relative quantization method 2^−ΔΔCT^. Each reaction was performed with three technical replicates and three biological replicates. 

### 4.6. Vector Construction and Genetic Transformation

To amplify *OsbZIP14* and *OsbZIP58* gene fragments, we used Nipponbare cDNA as the template and the Phanta Max Super-Fidelity DNA Polymerase P505 kit (Vazyme, Nanjing, China). The resulting products were ligated into the PMD-18T (TaKaRa) entry vector for sequencing (Tsingke Bio, Beijing, China). To construct various vectors, such as those for subcellular localization and yeast two-hybridization, we used the constructed entry vector PMD-18T as a template to amplify the corresponding sequences. We then used the Infusion Enzyme Linking System (TaKaRa) to connect the amplified fragments with the restriction vector. 

To construct the overexpression vector, a full-length cDNA of *OsbZIP14* was obtained by PCR using specific adaptor primers OE-OsbZIP14-F: 5′-GTCGACTCTAGAGGATCCACCATGGCATCGTCAAGCGGG-3′, and OE-OsbZIP14-R: 5′-TCGGGGAAATTCGAGCTCCTAGCAAAGCTGGAACAGCTCG-3′. The overexpression vector pUN-1301 was digested with the restriction enzyme BamHI and then linked with the Infusion enzyme (TaKaRa). The constructed vector was further verified and confirmed by sequencing (TsingKe). After the vector construction is completed, the rice callus was prepared, and the callus was inoculated in the agrobacterium culture medium for a total of 72 h at 20 °C. Screening medium was then carried out to obtain positive calluses and they were inoculated in differentiation medium. After differentiation of 2–5 cm buds, they were inoculated in a rooting medium and incubated at 30 °C for 7–10 days. Then the seedlings grew and rice genomic DNA was extracted by using the 2 × CTAB method to detect positive seedlings. Finally, the positive seedlings were transplanted to large fields for multiple generation propagation until the phenotype was stabilized.

The generation of knockout mutants for *OsbZIP14* in the ZH11 background was conducted via CRISPR/Cas9 gene editing as described by Wang et al.(2015) [56].

### 4.7. Subcellular Localization Analysis 

To determine the subcellular localization of the target gene products, rice protoplasts were prepared and transformed with plasmids using Coolaber’s Rice Protoplast Preparation and Transformation Kit (Beijing, China). The transformed protoplast cells were observed using a ZEISS LSM880 laser confocal system with an excitation wavelength of 488 nm and an absorption spectrum of 510–550 nm. For the subcellular localization of the native tobacco epidermis, activated Agrobacterium GV3101 containing the transformed plasmid was collected in an LB medium with an OD600 = 0.6. The cells were pelleted by centrifugation at 4000 rpm, resuspended in an injection solution consisting of 10 mM MgCl2, 10 mM MES (pH = 5.7), and 100μM AS (acetylsyringone), and then allowed to stand at room temperature for 3 h. The solution was injected into the 5-week-old tobacco leaf epidermis using a 1.0 mL syringe and spread to the entire blade. The injection site was cultured at 25 °C in the dark.

### 4.8. In-Silico Protein-Protein Interaction Analysis

Initially, we conducted the protein docking between *OsbZIP14* and *OsbZIP58* genes using the online HDOCK server (http://hdock.phys.hust.edu.cn/, accessed on 14 September 2022). Then from the result of the docking step, we selected the lowest energy model. Afterwards, the model was exported to PLIP (https://plip-tool.biotec.tu-dresden.de/plip-web/plip/index, accessed on 15 September 2022) and PDB-sum (http://www.ebi.ac.uk/thornton-srv/databases/pdbsum, accessed on 15 September 2022) for further analysis to identify the interacting amino acid residues.

### 4.9. Yeast Two-Hybrid (Y2H)

We generated fusion expression vectors BD-X using the in-fusion approach with pGBKT7 as the parent vector, following the instructions provided by the Matchmaker Gold Yeast Two-Hybrid System (Clontech 630489, Beijing, China). The transformed bacterial solution was spread onto a double-deficiency culture plate and incubated at 30 °C for 2–3 days.

### 4.10. BiFC Assay

To perform a BiFC assay, the fragments of *OsbZIP14* and *OsbZIP58* were fused to pRTVnVC and pRTVnVn plasmids, respectively, following a previously published method [57]. Rice protoplasts were prepared and transformed with the plasmids using the Coolaber’s Rice Protoplast Preparation and Transformation Kit. The fluorescence signals were detected using a ZEISS LSM880 confocal microscope.

### 4.11. DAB and Trypan Blue Staining

The clipped rice leaves were placed in a 10 mL centrifuge tube and then DAB and trypan blue staining solution were added to their respective centrifuge tubes at room temperature. Then the solution was protected from light and shaken at 200 rpm for 10 h. Afterwards the staining solution was removed and 100% alcohol was added, which then was treated in a water bath at 80 °C until the solution was transparent (if the color of the solution became darker then the alcohol was changed). Finally, pictures of treated leaves were taken.

## 5. Conclusions

In this research, we obtained the profiles of chromatin accessibility and gene-expression of rice under heat stress to demonstrate the chromatin structure alteration in the process of heat-stress response activation and differentiation. Using this combined approach, we predicted three TF genes, including OsbZIP14, OsMYB2, and OsHSF7. Further investigation on the predicted transcription factor OsbZIP14 was conducted through a series of physiological and biochemical studies to reveal its role as a key regulator in response to heat stress. Through comprehensive bioinformatics analysis, we demonstrated that OsbZIP14 contained a basic-leucine zipper domain and primarily functioned as a nuclear TF with transcriptional activation capability. Additionally, we also validated the role of this gene in rice growth under heat stress by using OsbZIP14 gene overexpression and knockout lines during early seedling and the grain-filling stages. Finally, by combining computational and experimental methods we verified the interaction between OsbZIP14 and OsbZIP58 gene, a key regulator of rice seed storage protein (SSP) accumulation. These findings provide good candidate genes for genetic improvement of rice and offer valuable scientific insights into the mechanism of heat tolerance stress in rice. 

## Figures and Tables

**Figure 1 ijms-24-05619-f001:**
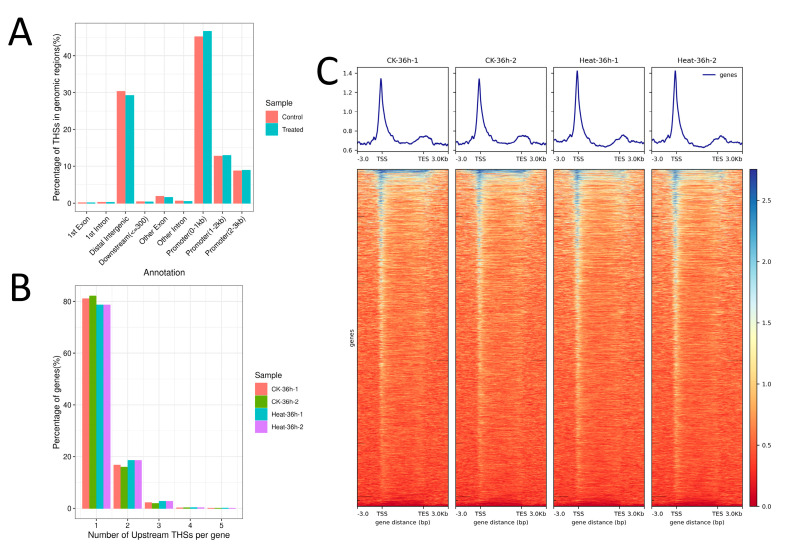
Distribution of THS in rice under heat stress. (**A**) THS genomic distribution in control and heat stress treatment (36 h). (**B**) The distribution of THSs in the upstream region of each rice gene. (**C**) The average plot and heatmap of ATAC-seq signals for over-enriched THSs in control (CK 36 h-1 and 2) and 36 h heat stress groups (Heat 36 h-1 and 2).

**Figure 2 ijms-24-05619-f002:**
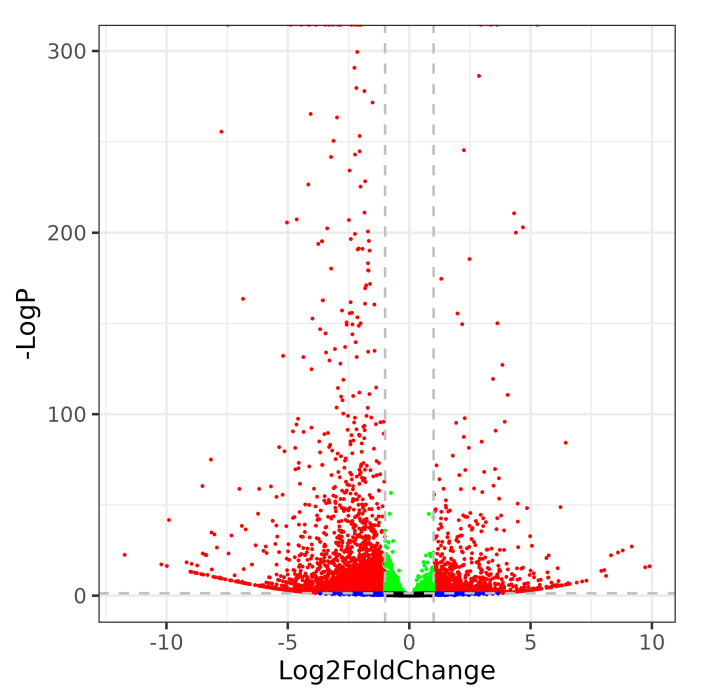
Distribution of DEGs in rice under heat stress. Red dots represent genes with |log2FC|≥1 and *p*-Value ≤ 0.05, green dots represent genes with |log2FC| < 1 and *p*-Value < 0.05, blue dots represent genes with |log2FC|>1 and *p*-Value > 0.05, and black dots for genes with no obvious statistical significance.

**Figure 3 ijms-24-05619-f003:**
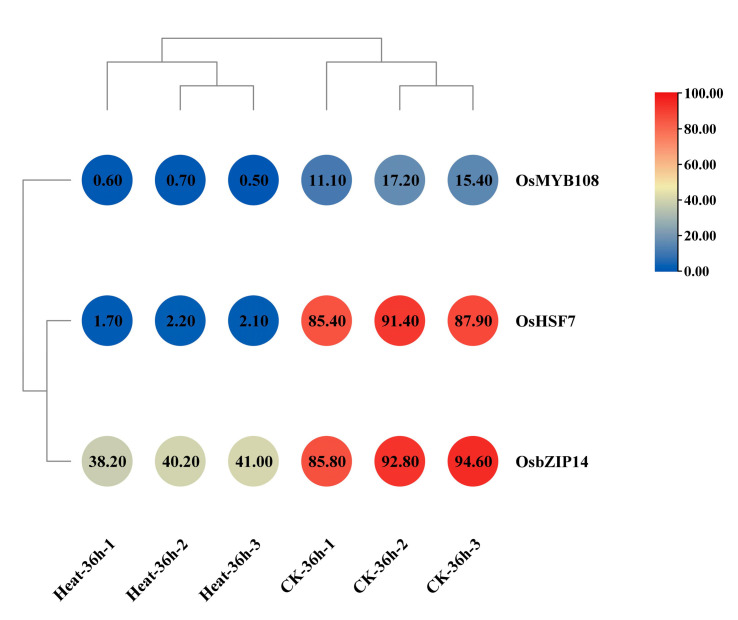
Heatmap of expression profiles of key TF genes in rice under heat stress. Z-scores were utilized to normalize the expression data of TF genes obtained from RNA-seq data analysis.

**Figure 4 ijms-24-05619-f004:**
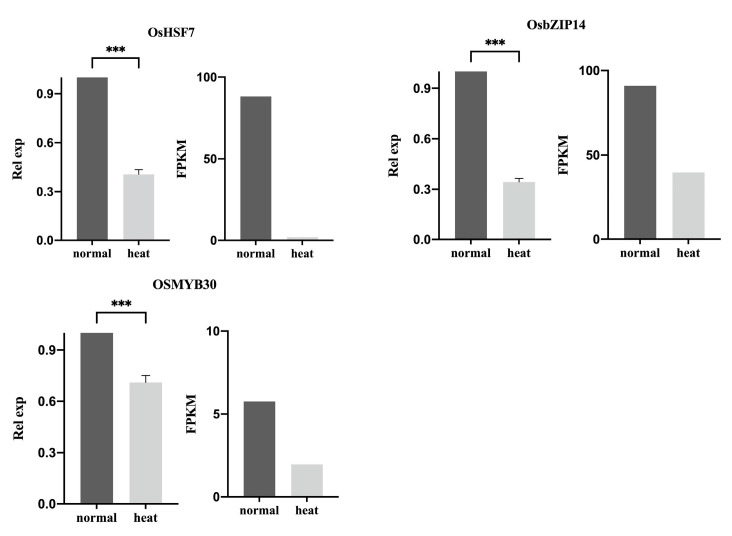
Expression of OsHSF7, OsbZIP14, and OsMYB30 by using RT-qPCR and RNA-seq methods. The plot on the left side represents the RT-qPCR measurement results, while the right side represents the RNA-seq analysis results. The statistical significance of the expression changes was determined using the student *t*-test, ‘***’ indicating a *p*-Value < 0.001.

**Figure 5 ijms-24-05619-f005:**
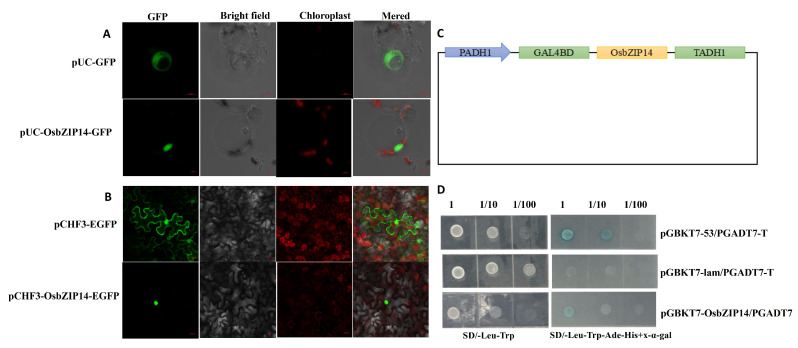
Subcellular localization and self-activation ability of OsbZIP14 gene. (**A**) The subcellular localization of OsbZIP14 protein in rice cells (Scale bar represents 5.0 µm). (**B**) Visualization of the subcellular localization of the OsbZIP14 protein in tobacco epidermal cells (Scale bar represents 20.0 µm). (**C**) Schematic representation of the PGBKT7-OsbZIP14 vector plasmid. (**D**) Assessment of the self-activation potential of the OsbZIP14 gene product protein.

**Figure 6 ijms-24-05619-f006:**
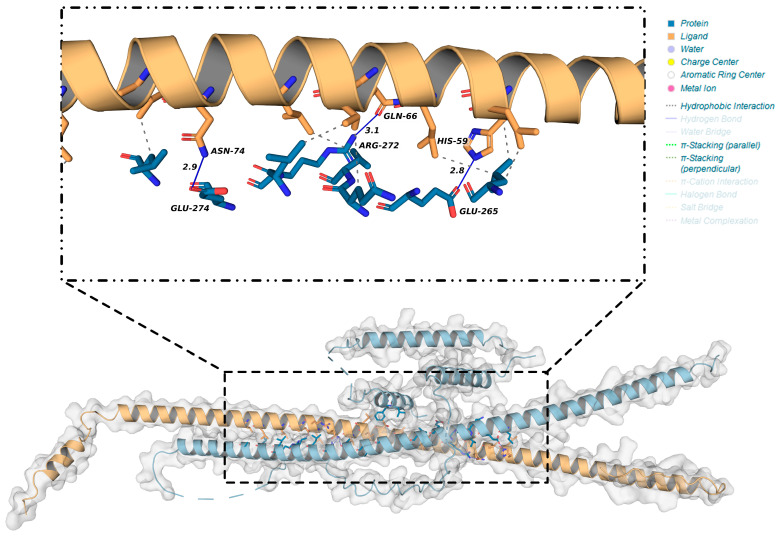
PLIP tool analysis result of the interaction between OsbZIP14 and OsbZIP58 products.

**Figure 7 ijms-24-05619-f007:**
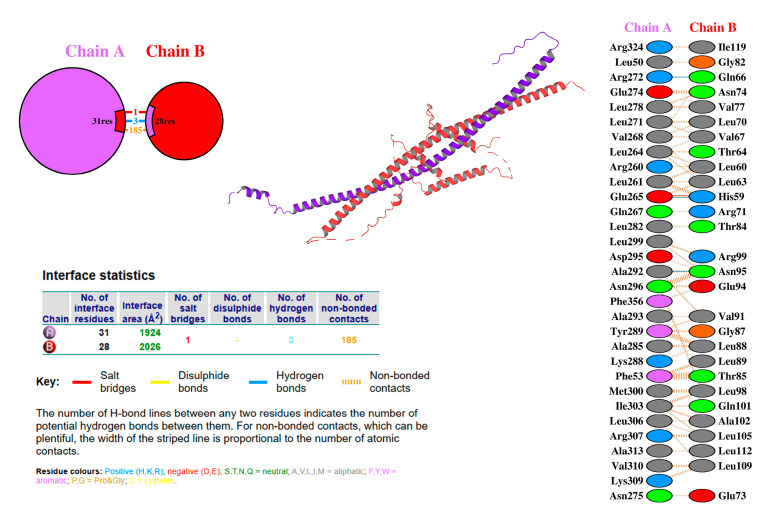
*PDBSum* tool *analysis result of the* interaction between OsbZIP14 and OsbZIP58 products.

**Figure 8 ijms-24-05619-f008:**
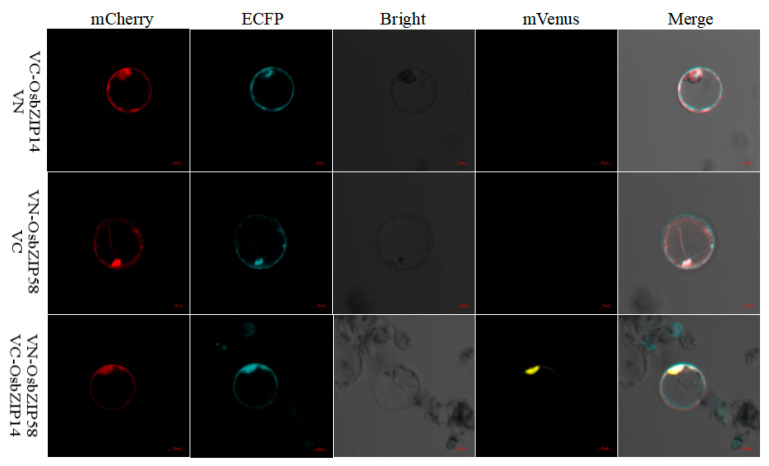
Detection of the interaction between OsbZIP14 and OsbZIP58 using BiFC. BiFC results showed a yellow fluorescence signal in rice protoplasts with the co-expression of the N-OsbZIP58/C-OsbZIP14, indicating a direct interaction between OsbZIP14 and OsbZIP58 (Scale bar represents 10.0 µm).

**Figure 9 ijms-24-05619-f009:**
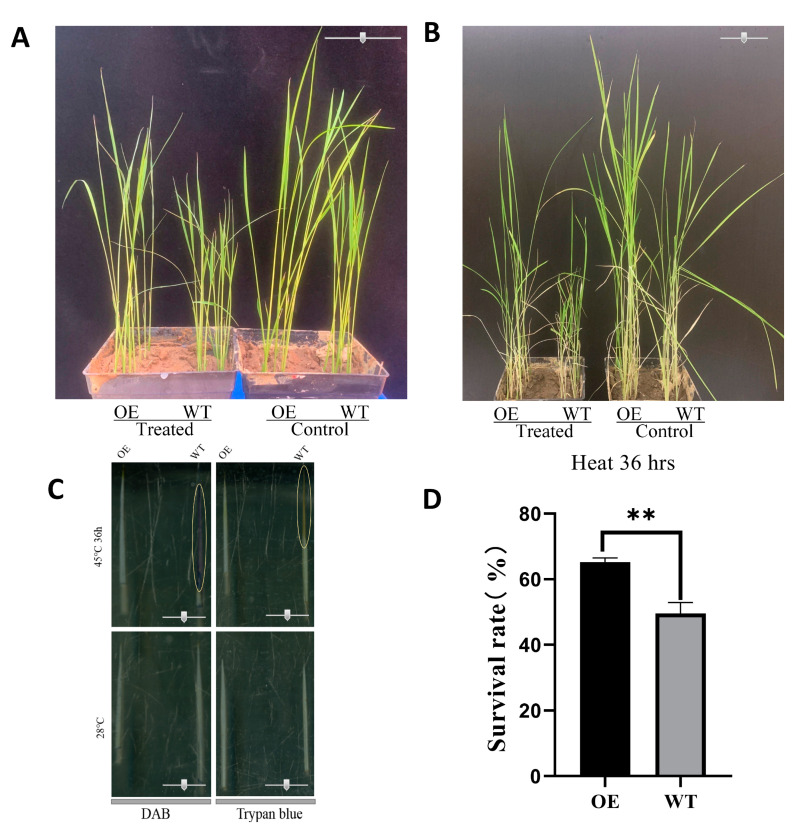
Effect of heat stress on the growth of OsbZIP14 OE and its WT rice seedlings. (**A**) Phenotypes of OE and WT seedlings under heat stress. Seedlings were grown in 28 °C soil for 9 days, then treated at 45 °C for 36 h (scale bars = 5.0cm). (**B**) Rice seedlings under 28 °C normal growth condition after one month of heat stress (Scale bars = 5.0 cm). (**C**) DAB staining of seedling leaf tips before and after heat stress treatment (Scale bar represents 2.0 cm). (**D**)The survival rate of the rice seedlings after one month of heat stress. (The statistical significance of the expression changes was determined using the student t-test, with ‘**’ indicating a *p*-Value < 0.01).

**Figure 10 ijms-24-05619-f010:**
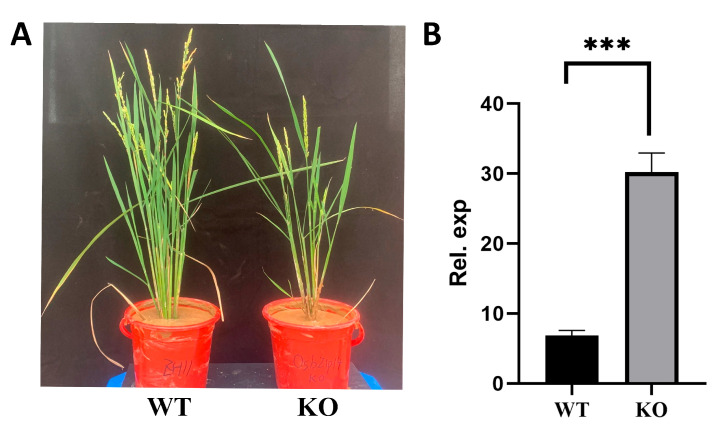
OsbZIP14 gene knockout and its WT plants under heat stress. (**A**) Overall phenotypic comparison of WT and OsbZIP14 KO rice plants under 5d heat stress. (**B**) Analysis of OsbZIP58 expression levels in WT and mutant rice plants under 5d heat stress. The statistical significance of the expression changes was determined using the student *t*-test, ‘***’ indicating a *p*-Value < 0.001.

**Table 1 ijms-24-05619-t001:** Heat-specific enriched motifs and their binding TFs in rice.

Enriched Motif	TF (Arabidopsis)	TF (Rice)	Gene ID
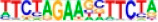	HSFA1E	OsHSF7	Os03g0161900
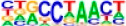	MYB108	OsMYB2	Os12g0175400
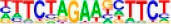	HSF3	OsHsfB2b	Os08g0546800
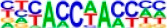	MYB63	OsMYB30	Os02g0624300
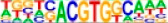	bZIP2	OsbZIP14	Os02g0132500

**Table 2 ijms-24-05619-t002:** Expression abundance and binding sites of heat-responsive TFs in rice.

Enriched Motif	TF (Arabidopsis)	TF (Rice)	Gene ID	Log2FC (Nip)	FDR	*p*-Value
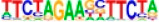	HSFA1E	OsHSF7	Os03g0161900	−5.48	0.0000	1.00 × 10^−18^
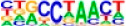	MYB108	OsMYB2	Os12g0175400	−4.65	0.0000	1.00 × 10^−6^
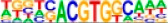	bZIP2	OsbZIP14	Os02g0132500	−1.20	0.0000	1.00 × 10^−7^

**Table 3 ijms-24-05619-t003:** The top 10 docking models between *OsbZIP14* and OsbZIP58.

Rank	Model1	Model2	Model3	Model4	Model5	Model6	Model7	Model8	Model9	Model10
Docking Score	−441.33	−436.81	−378.22	−350.78	−345.74	−345.02	−336.86	−330.27	−326.05	−324.79
Confidence Score	0.9971	0.9968	0.9897	0.9823	0.9804	0.9802	0.9767	0.9735	0.9713	0.9706
Ligand rmsd (Å)	120.03	122.27	24.74	119.18	26.21	25.07	38.82	50.8	22.74	119.18

## Data Availability

The data presented in this study are available in article or Appendix A here.

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
