# Peer review of "Integrated ATAC-Seq and RNA-Seq Data Analysis to Reveal OsbZIP14 Function in Rice in Response to Heat Stress"

_ijms, 2023, doi:10.3390/ijms24065619_

Round 1
Reviewer 1 Report
The manuscript require major revision, and must be revised as discussed below-
1. The authors mentioned that RNA-seq data were retrieved from the NCBI. Authors must provide information about the rice genotype used for each SRA study and how the polymorphism among the genotypes was treated.
2. How were the OsbZIP14 over-expression (OE) and OsbZIP14 knockout (Ko) lines developed?
3. The quality of the result images is too poor to be looked at and understood. I strongly advise authors to improve the figure quality to clarify the axis and axis legends.
4. Were there some common motifs among KO and OE lines?
5. Authors confirmed the transcriptional activity in the OsbZIP gene and performed the modeling. what was the docking protein used to interact with OsbZIP? again the docking results are not clear and have very low resolution. please improve.
6. How many heat-responsive TFs were identified? A list of the TFs should be supplied.
Author Response
Dear Reviewer 1,
Thank you for giving us the opportunity to submit a revised draft of the manuscript “Integrated ATAC-seq and RNA-seq data analysis to reveal OsbZIP14 function in rice in response to heat stress” for publication in the International Journal of Molecular Sciences. We appreciate the time and effort that you and the reviewers dedicated to providing feedback on our manuscript and are grateful for the insightful comments on and valuable improvements to our paper. We have incorporated most of the suggestions made by the reviewers. All the changes made to the revised manuscript file are highlighted by using tracked changes.
Point 1. The authors mentioned that RNA-seq data were retrieved from the NCBI. Authors must provide information about the rice genotype used for each SRA study and how the polymorphism among the genotypes was treated.
Response 1: The rice genotype used in both the RNA-seq and ATAC-seq data analysis is Nipponbare rice which is the same to the rice reference genome. As a result we didn’t need identify the polymorphic sites in the samples used. We also updated the names of genotypes in the materials and methods section.
Point 2. How were the OsbZIP14 over-expression (OE) and OsbZIP14 knockout (Ko) lines developed?
Response 2: The development of both the over-expression(OE) and knockout(KO) is explained in methods and materials section, Vector Construction and genetic transformation sub-section.
Point 3. The quality of the result images is too poor to be looked at and understood. I strongly advise authors to improve the figure quality to clarify the axis and axis legends.
Response 3: We apologize for the poor quality images but in this revised version we improved the resolution of all the images.
Point 4. Were there some common motifs among KO and OE lines?
Response 4: Although we didn’t verify the common motifs between the KO and OE lines, given the fact that both the OE and KO lines belong to the same family of rice species and we initially verified that there are no polymorphic sites in the OsbZIP14 gene between ZH11 and Nipponbare rice variety, there is highly likely that there are common motifs among these lines.
Point 5. Authors confirmed the transcriptional activity in the OsbZIP gene and performed the modeling. what was the docking protein used to interact with OsbZIP? again the docking results are not clear and have very low resolution. please improve.
Response 5: Thanks for pointing it out and we again apologize for not providing good quality images. We used the OsbZIP14 gene to interact with OsbZIP58 and selected the model with the lowest energy. We also improved the image quality and we splitted image into two separate figures so that it can be visualized better(Figure 6 and Figure 7).
Point 6. How many heat-responsive TFs were identified? A list of the TFs should be supplied.
Response 6: We apologize for not providing the whole list of rice TF genes. After we blasted the rice genes against known Arabidopsis thaliana with highly enriched TFs, overall we identified 37 TF rice genes(e-value<1e-5) which were highly similar to 8 Arabidopsis thaliana (Provided as a supplementary material Table S2). We then further selected the five rice TF genes with higher identity to the Arabidopsis thaliana TFs for additional analysis.
Reviewer 2 Report
ijms-2277975-peer-review-v1
The manuscript ‘Integrated ATAC-seq and RNA-seq data analysis to reveal OsbZIP14 function in rice in response to heat stress’ by Qiu et al. is a well-planned, well-executed, and well-written; however, I have a few suggestions which could improve the manuscript.
Although the MS is well-written, I suggest the authors improve the language and punctuation. Please improve the grammar score throughout the MS. I am citing few examples;
IN ABSTRACT, ‘Transcription factors (TFs) play critical roles in mediating the plant response to various abiotic stresses, particularly in the context of heat stress.’ Could be written as ‘Transcription factors (TFs) are critical in mediating the plant response to various abiotic stresses, particularly heat stress’.
The sentence “Despite their immense importance, a limited number of heat-stress-responsive TFs have been identified in rice to date, and the molecular mechanisms underpinning the role of TFs in rice adaptation to heat stress remain largely inconclusive.” May be revised as
“Despite their immense importance, a few heat-stress-responsive TFs have been identified in rice. The molecular mechanisms underpinning the role of TFs in rice adaptation to heat stress still need to be more conclusive.”
Please re-write the sentence “In this study, three TF genes, including OsbZIP14, OsMYB2, and OsHSF7, were identified by integrating transcriptomic and epigenetic sequencing data analysis of rice in response to heat stress.” clearly as “his study identified three TF genes, including OsbZIP14, OsMYB2, and OsHSF7, by integrating transcriptomic and epigenetic sequencing data analysis of rice in response to heat stress.”
Please mention all the genes, TFs in italics throughout the MS.
Please mention ‘the’ before the particular gene or TFs, example; knocking out ‘the’ OsbZIP14 (Line 24); expression of ‘the’ OsbZIP58 gene (Line 27).
Line 24: Please delete ‘Notably,’
Line 26: ‘High-temperature’
Line 29: Please delete ‘Collectively,’
The sentence ‘These findings not only provide promising candidate genes for genetic improvement of rice but also offer valuable scientific insights into the mechanism of heat tolerance stress in rice.’ May be revised as ‘These findings provide good candidate genes for the genetic improvement of rice but also offer valuable scientific insights into the mechanism of heat tolerance stress in rice.’
Please use either ‘Rice’ or ‘Oryza sativa L.’ in the keywords.
INTRODUCTION
Line 42: “Heat has adverse effects on rice quality” may be revised as “Heat adversely affects rice quality”.
Line 49-50: ‘The identification of’ may be revised as ‘identifying’
Line 54: Please delete ‘thereby’
Line 56: Please delete ‘At present,’
Line 58: ‘due of’ or ‘due to’? please revise
Line 63-67: May be revised as ‘However, as high-throughput sequencing technology advances, many omics technologies emerge. Any mono-omics is insufficient to explain the mechanism of the plant stress response systematically, and the combination of multi-omics is an inevitable trend of future development [21].’
Line 78: Please revise ‘the rapid and efficient identification of’ as ‘rapidly and efficiently identifying’.
Line 80-81: Please revise the sentence as ‘To our knowledge, only a few TF genes have been reported that play critical roles in rice in response to heat stress.’
Line 84: Please revise ‘but also provide’ to ‘and provide’.
Line 107-108: Please check the legend to figure. What is ‘n’?
Results are well-presented.
Line 224: Please revise “The yellow light in”
Line 233-245: ROS detection using DAB and lipid peroxidation procedures using trypan blue staining are missing in methods. Please update.
Line 258: ‘osbzip14’ …please write the gene name properly
The results are well-discussed.
Line 286: Liu et al. Please revise ‘et. al.’ throughout the MS as ‘et al.’
Line 309: Please mention the scientific name in italics ‘Arabidopsis thaliana’
Discussion on protein docking, ROS, Lipid peroxidation, could add more value to the section.
Materials and methods:
Please mention the date of assess of all the software used.
Line 345: What is padj???
Line 354: Please write the version of R. What is R package Diffbind?
Line 375: ‘TaKaRa’…Please write the city and country of the consumables/company used in the MS.
Line 381: Please check the sentence case in the headings and sub-headings.
Line 384: ‘TAKARA’: Please maintain coherence (Line 375 and 388).
Line 412: ZEISS or ZESS? Please check
Please mention the docking methods in the introduction, materials and method, and discussion sections (as mentioned in the result).
Please mention whether any admit screening of genes has been performed before protein-protein docking? Please mention the detailed procedure.
Line 414-421: Please re-write the conclusion with the key findings and prospects. Graphical representation of the future scope may not be required, and please be deleted.
References may be arranged as per the journal pattern. Please cross-check the references cited with the list.
Language, punctuation, and grammar may be cross-checked.
Good luck with the revision.
Author Response
Dear Reveiwer 2
Thank you for giving us the opportunity to submit a revised draft of the manuscript “Integrated ATAC-seq and RNA-seq data analysis to reveal OsbZIP14 function in rice in response to heat stress” for publication in the International Journal of Molecular Sciences. We appreciate the time and effort that you and the reviewers dedicated to providing feedback on our manuscript and are grateful for the insightful comments on and valuable improvements to our paper. We have incorporated most of the suggestions made by the reviewers. All the changes made to the revised manuscript file are highlighted by using tracking changes function.
Point 1. Suggestions to rewrite parts of the abstract and Introduction sections
Response 1: Thanks again for your valuable suggestions, we made all the changes according to your input. The changes are highlighted using the tracking changes function in microsoft word.
Point 2: Line 233-245: ROS detection using DAB and lipid peroxidation procedures using trypan blue staining are missing in methods. Please update.
Response 2: We updated in the method and materials section , subsection 4.11.
Point 3. Discussion on protein docking, ROS, Lipid peroxidation, could add more value to the section.
Response 3: Thanks again for your important input. We revised according to your suggestion. The changes we made are, for the protein docking part – Line 327-339 and for the ROS, Lipid perodixation part – Line 308-319
Point 4. Materials and methods: Please mention the date of assess of all the software used.
Response 4: We included all the access dates for the softwares and website we used.
Point 5: Line 286: Liu et al. Please revise ‘et. al.’ throughout the MS as ‘et al.’, Please mention the scientific name in italics ‘Arabidopsis thaliana’,
Response 5: All the changes are made accordingly.
Point 6. Line 345: What is padj???
Response 6: Padj is short for adjusted p-value or according to our manuscript it means the corrected p-value
Point 7: Line 354: Please write the version of R. What is R package Diffbind?
Response 7: We apologize for not writing it correctly, we meant Diffbind package in R environment . We included the version of R used in this revised version.
Point 8: Line 375: ‘TaKaRa’…Please write the city and country of the consumables/company used in the MS., Line 381: Please check the sentence case in the headings and sub-headings. ,Line 384: ‘TAKARA’: Please maintain coherence (Line 375 and 388). ,Line 412: ZEISS or ZESS? Please check
Response8: All are corrected according to your suggestion.
Point 9. Please mention the docking methods in the introduction, materials and method, and discussion sections (as mentioned in the result).
Response9: We included the docking methods in all the mentioned sections.
Point 10: Please mention whether any admit screening of genes has been performed before protein-protein docking? Please mention the detailed procedure.
Response 11: we didn’t do any admit screening.
Point 12: Line 414-421: Please re-write the conclusion with the key findings and prospects. Graphical representation of the future scope may not be required, and please be deleted.
Response 12: Thanks for your suggestion again, we revised and rewritten the conclusion and removed the figure too.
Point 13: References may be arranged as per the journal pattern. Please cross-check the references cited with the list., Language, punctuation, and grammar may be cross-checked.
Response 13: Thanks for your valuable comment again, we revised the English language of the whole manuscript and we also formatted the reference citation and bibliography according to IJMS style.